# Pandemic and Economy: An Econometric Analysis Investigating the Impact of COVID-19 on the Global Tourism Market

**Ioannis-Panagiotis Varzakas and Theodore Metaxas ***

Department of Economics, University of Thessaly, 38333 Volos, Greece; varzakas@uth.gr
* Correspondence: metaxas@uth.gr

**Abstract:** With the outbreak of the COVID-19 pandemic, the global tourism market has become one of the most affected sectors of the economy. In this research, the literature on the economic effects created by COVID-19 on a global level is first studied and the measures and restrictions that governments are obliged to take in order to suppress and prevent the spread of the coronavirus are analyzed. Next, there is an attempt to empirically estimate a system of equations regarding the two channels of influence of COVID-19 on tourism, making use of cross-sectional data, and specifically for a sample of 38 countries that launched vaccination up until the end of 2020. The explained results confirm that tourism is directly affected by the spread of COVID-19, due to the effort of travelers to avoid illness, but also due to the measures taken by governments to limit it. Tourism is also indirectly affected, due to the negative impact on income. Using 3SLS, an equation was developed to calculate the direct and indirect impact of COVID-19 on tourism. Notably, the dependent variable (tourism expenditure) exhibited a positive correlation with the independent variable (GDP) and a negative correlation with the total COVID-19 cases. Consequently, it was determined that a unit increase in the COVID-19 variable led to a reduction in international tourism expenditure by USD 859,237. Finally, there is a concern, and further investigation is needed, regarding the effect of vaccination against COVID-19 on tourism, which, while it is expected to be negative, is not confirmed by the results.

**Keywords:** COVID-19; global tourism market; econometric analysis

## 1. Introduction

The tourism industry is one of the most important and fastest growing sectors of the global economy, generating profits and jobs. However, in December 2019, the COVID-19 virus appeared. This particular coronavirus is observed to be highly contagious and deadly, wreaking havoc on the travel and tourism industry. The world experienced an unprecedented situation. Governments around the world implemented drastic measures to suppress the spread of COVID-19. The outbreak of the pandemic forced many businesses, especially in tourist-dependent destinations, to partially or completely stop their activities, causing huge economic costs as well as millions of jobs being lost. In addition to the human suffering and millions of deaths that COVID-19 has caused, there is also a huge economic impact, the duration and depth of which is difficult to predict.

The purpose of this research is to investigate the negative effects of COVID-19 on the global tourism market for the year 2020–2021. In the theoretical part of this study, a bibliographic review is carried out, analyzing the situation that prevails in the tourism industry. An econometric analysis follows, in which the economic impacts on the tourism industry are investigated and compared using various variables. What this research has to add are the results of the impact of COVID-19 on the tourism industry globally, through an econometric analysis for the year 2020. Most of the econometric studies carried out so far investigate a specific region. In the analysis that follows, it has been taken into account how vaccination affects tourism, which has not been investigated as a variable. Moreover,

the system of equations that was developed can be used in further analysis of various crisis events, estimating the economic impact by examining different variables.

In the second section there is a historical review of the development of the tourism industry from 1950 to 2019, followed by a brief description of the first shock that the world suffered with the spread of COVID-19, while in the third a comparison is made with other crisis events and how the world tourism industry was affected. In section fourth, extensive reference is made to vaccination, as well as to the repressive and preventive measures used by governments to reduce the spread of COVID-19. This is followed by a literature review, in Section 5, analyzing the economic effects of the pandemic in the world and how tourism indicators were affected. In the sixth section, reference is made to previous studies analysis. The following seventh section reports and analyzes the meanings used as variables for the econometric analysis. The methodology used for this research is also analyzed. In the last section, Section 8, the main conclusions of the overall study carried out are mentioned.

## 2. The Global Tourism Market until 2019, in Brief

Tourism is the sector that is based on the need of people to travel to new places; in addition, tourism is often combined with activities other than pleasure [1]. Moreover, tourism has a critical role in the development of economies worldwide [2]. Its importance can be identified by the fact that it has been included in three of the 17 Sustainable Development Goals declared by the United Nations [3]. Tourism, as we know it today, according to many historians, began during the Industrial Revolution in England with the rise of the middle class and the liberalization of international air transport, as well as the rise of low-cost airline companies [4].

After the end of the Second World War, spectacular growth of the tourism industry followed, resulting in the global tourism market being one of the largest and most profitable industries worldwide. According to the UNWTO, from 1950 to 1990 international arrivals increased by 440 million travelers, reaching its peak at 2019, with travelers exceeding 1.5 billion. By 2019, over 320 million people worldwide were employed in the tourism industry, meaning that 1 in 10 jobs belonged to the tourism sector. In fact, tourism is one of the fastest growing industries, providing approximately 10% of employment nationally and accounting for 10% of global GDP [5]. The development of the global tourism sector in the last 70 years did not proceed smoothly, but included declines and recoveries. However, the global tourism market still ranks fourth in the category of exports.

## 3. The First Shock

The tourism industry has been affected in recent years by an increasing number of crises and disasters [6,7]. Although there are various opinions about crisis management on the tourism industry and the knowledge that exists about it [7,8], the current COVID-19 pandemic is perceived as a crisis of unprecedented magnitude, especially in the light of its effects globally [9].

The global tourism market is one of the most susceptible sectors of the economy in terms of crises, whether natural or human disasters [10]. The recovery period is longer, compared to other sectors of the economy, specifically if a place is closely linked to a crisis or a disaster [10,11].

COVID-19 originated from a deadly infection (SARS-CoV-2) in Asia [12]. This virus was first discovered in Wuhan by health experts at the end of December 2019. The World Health Organization (WHO) declared the COVID-19 outbreak as a public health emergency of international concern in January 2020. By mid-February 2020, the virus had managed to be transmitted to 146 countries via global air transport. In March 2020, it was categorized as a pandemic. What made the situation unique was the unprecedented use of large-scale restrictive measures like curfew, various types of lockdowns, closure of shops and hotels and restrictions on the number of people in closed spaces.

Regarding the global tourism market, COVID-19 is not the first risk in history to which this industry has been exposed. The tourism industry has faced similar risks and crises in the past.

According to the World Bank [13], it is widely known that during the 20th century humanity experienced three major crisis events. These were the Spanish flu of 1918–19, the Asian flu of 1957 and the Hong Kong flu of 1968. However, international arrivals were not affected at all.

As for the 21st century, in a span of just fifteen years, from 2000 to 2015, major disruptive events took place, such as the terrorist attack of 9/11, SARS and its outbreak in 2003, the global financial crisis in 2009, the Ebola virus in 2014 and MERS in 2015 [13]. Figure 1 presents how crisis events affect international tourism arrivals.

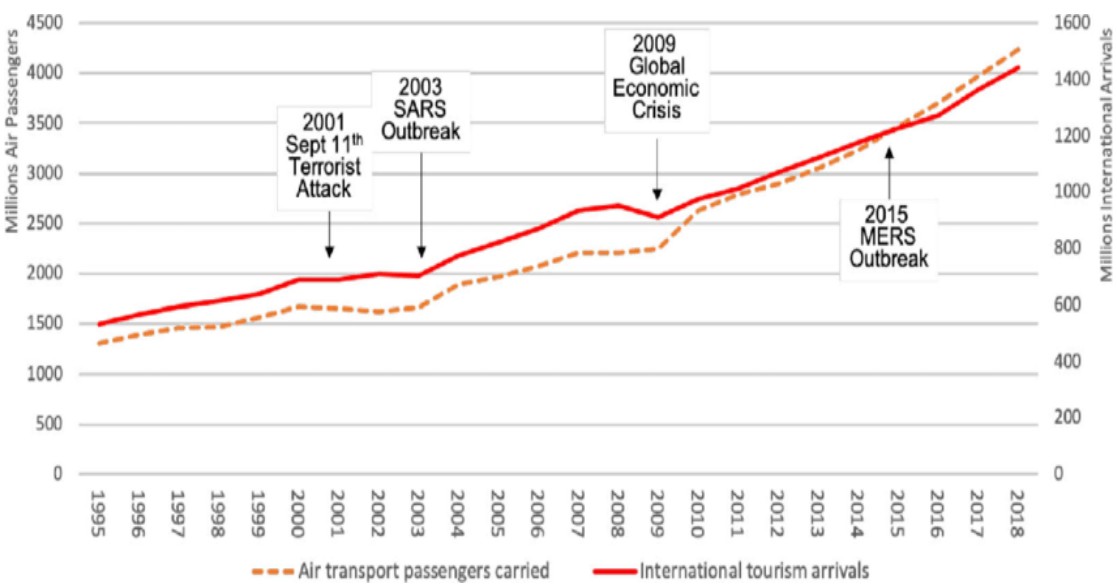

**Figure 1.** How crisis events affect the number of air-transport passengers carried and international tourism arrivals. Source: World Bank [14,15].

However, none of the crises described led to long-term negative effects on the tourism sector. Only SARS (2003) and the global financial crisis (2008–2009) had the effect of reducing international arrivals by 0.4% and 4%, respectively.

In addition, a lot of research has been carried out in the tourism industry on the possible systemic effects of global climate change. However, there has not been the same interest regarding the effects on the world tourist market, taking into account the possibility of the outbreak of a pandemic. Most research to date has focused on individual countries or regions.

Several studies have demonstrated that the role of air travel is important in the spread of infectious diseases such as influenza, as well as various groups of coronaviruses [16].

## 4. Public Health Strategies to Control COVID-19

The World Health Organization [17] defines herd immunity, also known as population immunity, as the indirect protection against an infectious disease that occurs when a population acquires immunity either through vaccination or immunity developed through previous infection. During the outbreak of COVID-19 and before the vaccination, the medical community and the public health sector had only non-pharmaceutical interventions against COVID-19. Their ultimate goal was to mitigate the pandemic using these tools. The specific category of measures, non-pharmacological interventions (NPIs), were aimed at suppressing the peaks of the COVID-19-outbreak waves, smoothing the curve and delaying the resurgence of the virus. In this way, the smooth operation of the public health system

of each country was ensured, as well as the reduction of morbidity and mortality caused by COVID-19 (Figure 2).

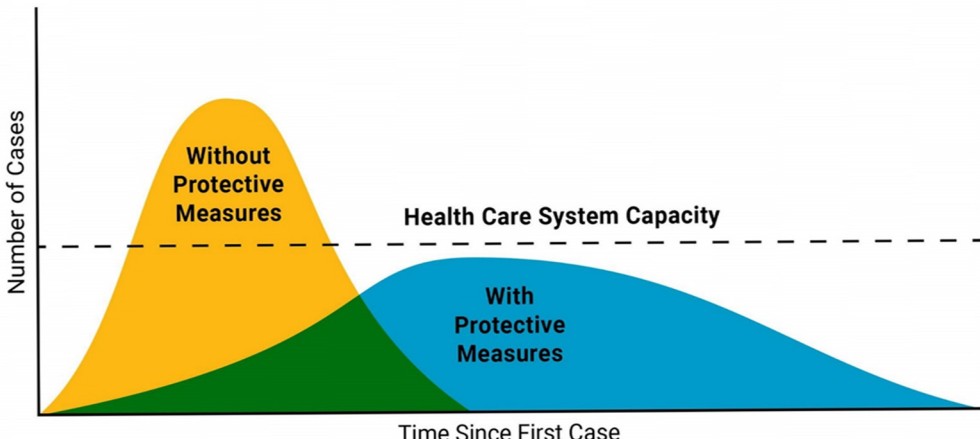

**Figure 2.** Curve of cases with and without the use of NPIs. Source: CDC [18].

Travel restrictions, as well as interventions on the physical distance, were implemented in many countries where there had been an outbreak and spread of the COVID-19. The implementation of travel restrictions, as well as social distancing measures, along with other interventions, such as mandatory tests (rapid, PCR) in a large part of the population, are likely to contribute to reducing rates of disease transmission and to flattening the curve [19].

The World Health Organization created several guidelines to support governments worldwide in planning and responding to the pandemic [17]. Other organizations such as ECDC and CDC published guidelines for different settings, including educational institutions, workplaces, communities, etc. These guidelines were based on experience from other epidemics, such as seasonal influenza and Ebola [17,20]. These guidelines detail the various scenarios regarding the health status of an area and the magnitude of the spread of COVID-19 in a given period. According to the WHO [17], the combination of these measures is even more effective than individual measures. These methods have been called "Defense in Depth" or "Multilevel Interventions".

Based on the toolkit prepared by ECDC, the main tools that governments could use fall into the following categories:

- Travel measures aimed at limiting the transmission of the virus from external sources (incoming cases).
- Personal protection measures aimed at limiting the possibility of virus infection for people operating and working in a high-risk environment.
- Social distancing measures aimed to eliminate the mobility of infected people and the spread of the virus in the general population.
- Antiviral drugs.
- Vaccines.

More specifically, regarding vaccination, on 8 December 2020, the first COVID-19 vaccine was delivered in the UK. According to Dube [21] about 60% of the world's population had received some kind of vaccine against COVID-19 up until March 2021. However, he observes that vaccination rates in developing countries amounted to 10%, far below the global average. With the peak of the COVID-19 waves, while many developing countries were fighting against the coronavirus, it is observed that in the developed countries of Europe and North America, business and social life were partially returning to normal levels, due to a high vaccination rate. The reboot of the economy in the northern developed countries also meant that these regions could redirect their economic resources for the purpose of reconstruction. On the other hand, developing countries continued to spend huge amounts of money on health and society, with the possible result of worsening

prospects for economic development in these countries, including the tourism sector. This result, according to Dube [21], is the great inequity that existed regarding the availability of vaccines in developing countries (Figure 3).

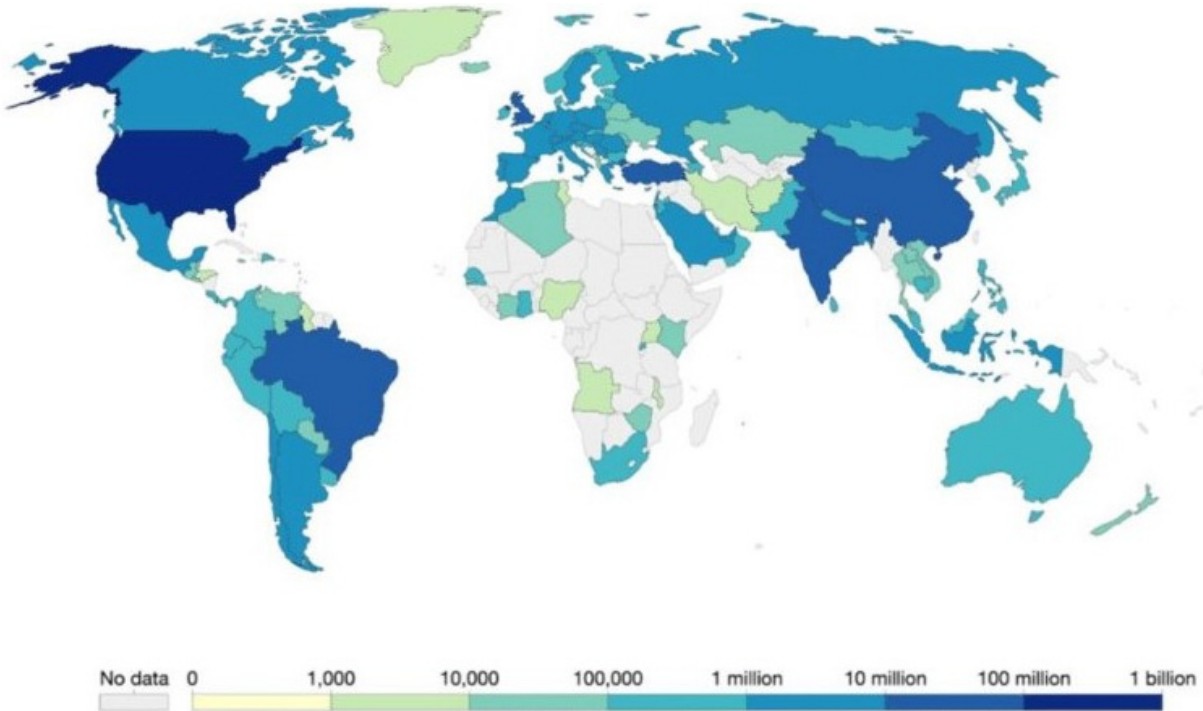

**Figure 3.** Total number of installments allocated worldwide as of 17 March 2021. Source: Our World in Data [22].

According to the WHO [17], regarding the importance of vaccination, investing in preventive health will have positive outcomes for individuals, health systems and the wider economy. In particular, a 1% increase in vaccination rates, supported by public funding, is associated with a 0.6% increase in average annual expenditure by people aged 60 and over. Investing in vaccination can support leisure and travel spending, as the population ages. There are huge economic opportunities, but health is essential to realizing this potential increase in spending. People over the age of 65 represent a large share of the population, and seniors spend more than they used to in the past. People who are retired but still in good health spend a particularly large portion of their income and savings on leisure and travel.

## 5. Economic Impact of COVID-19 on the Global Tourism Market

The COVID-19 pandemic caused a significant blow to the global economy [23,24]. At the beginning of March 2020, the novel coronavirus was declared as a pandemic by the WHO, and since then a disturbance in the world markets has started [21]. Deaton [25] argues that COVID-19 led to income shocks which had a multiplier effect on tourism activities. Undoubtedly, tourism is the sector most affected by COVID-19, worldwide.

With the declaration of COVID-19 as a pandemic, the global market and supply chain was disrupted due to closed borders, as various countries established lockdowns. Moreover, tourists were considered as potential carriers of COVID-19 [26]. Supply and distribution of products worldwide was disrupted in most industries, excluding commodities. This resulted in the creation of a huge imbalance in global demand and supply. The effects of the imposed restrictions were initially estimated to affect only the manufacturing sector, but over time they further extended to the services sector, as well [27]. It was also observed that during the peak of the first two waves of COVID-19, in 2020, there was a suspension of

activities in airlines, large hotel units, restaurants, car rental companies, cruise ships and national parks [21].

According to the UNWTO (2021), it is revealed that in 2020 the global travel and tourism industry lost approximately USD 4.5 trillion in global GDP, with the sector shrinking by 49.1% against a global economic shrinkage of 3.1%. As a consequence of this crisis, there has been a huge impact on the global labor market, as well as on tourism jobs, since COVID-19 restrictions led to 62 million people losing their jobs. While the initial effects of the pandemic across the industry could be equable, recovery is expected to be incremental, fraught with challenges that vary across geographic regions [28].

Even specifically, regarding tourism industry indicators during COVID-19, according to the IATA [29], the COVID-19 pandemic has led to the biggest shock to the air travel and the airline industry since World War II. COVID-19 had a direct effect on air transport. The negative impact the pandemic has had on the aviation industry for 2020 is a dramatic 66% decrease, measured in passenger kilometers (RPKs), compared to 2019 [30] (Figure 4).

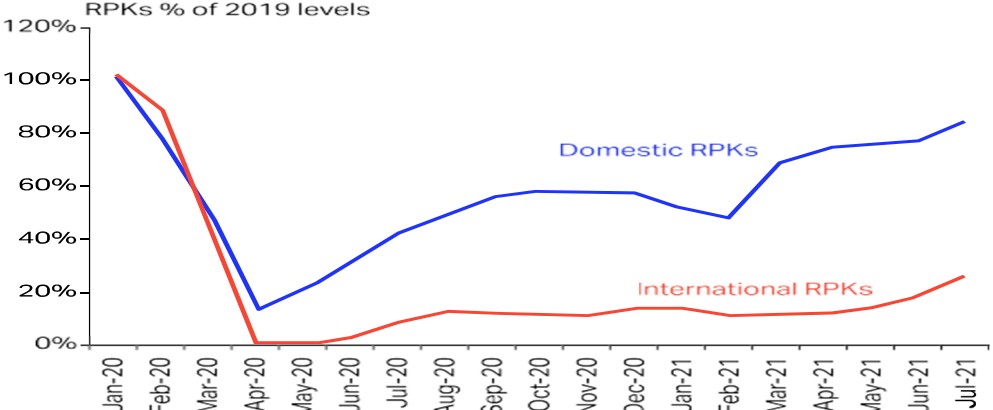

**Figure 4.** RPKs for domestic and international arrivals with a base year of 2019. Source: IATA Economics [29].

After the closure of the borders, there followed the cancellation and postponement of major events, as happened with the Olympic Games in Tokyo. The restrictions implemented by many countries also had a devastating impact on the car rental, hotel and tourism markets. Another sector that was significantly affected by the spread of the pandemic was the hotel industry.

The hotel industry was constantly growing, with more than 700,000 hotels worldwide, contributing over USD 3.41 trillion to the global economy [31]. With the outbreak of COVID-19, hotel bookings fell into historic lows; for 2020, occupancy amounted to 43%, i.e., it was reduced by 33.3% compared to 2019 [31]. According to the World Tourism Organization [32], following a 64% drop in international tourism receipts in 2020, global destinations recorded modest improvement in the first nine to eleven months of 2021, especially in Europe, although profits remained well below pre-pandemic 2019 levels. For the year 2020, international tourist receipts amounted to USD 546 billion.

As can be seen in Figure 5, international tourism receipts increased by 4% in 2021 compared to 2020, in real terms (local currencies, constant prices), but remained 62% below 2019. This rate is slightly better than that of international arrivals (−71% compared to 2019), due to a significant increase in expenditure per international trip during the pandemic [32]. Although there has been a slight improvement in tourism receipts worldwide and a significant recovery for domestic tourism, the direct gross domestic product (TDGDP) of tourism was projected for 2021 to be at around USD 1.9 trillion, while in 2020 it was USD 1.6 trillion. However, the results for these two years are really low in comparison with the 3.5 trillion in TDGDP recorded in the year 2019.

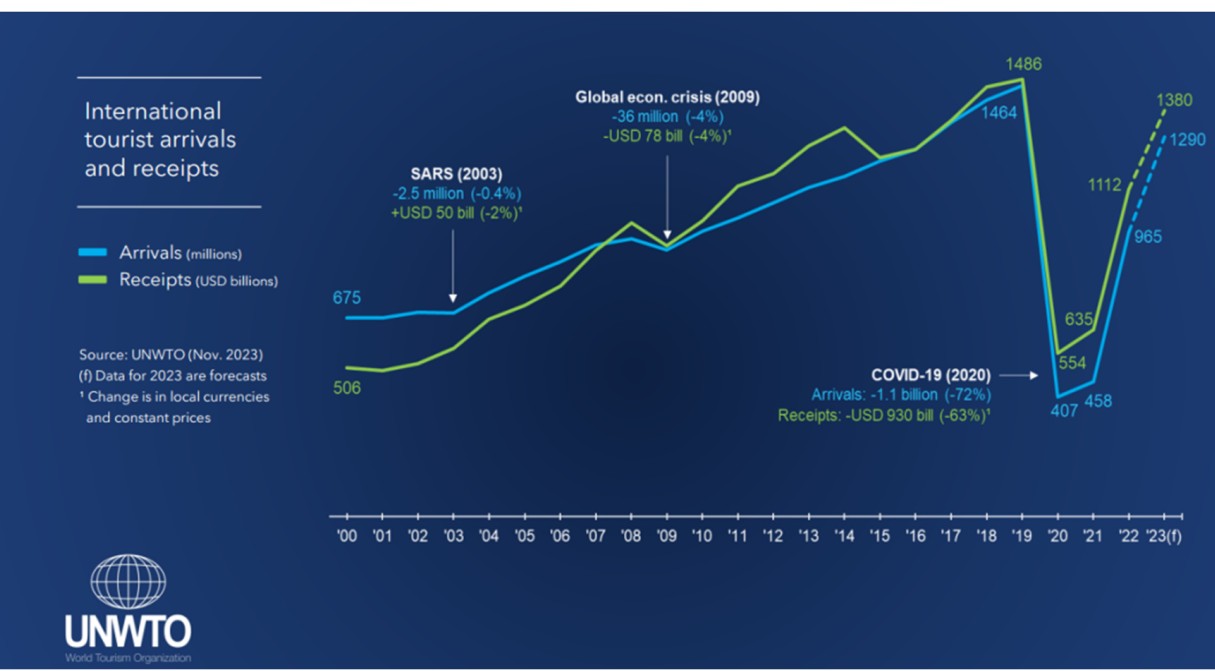

**Figure 5.** International Arrivals and Tourist Receipts. Source: UNWTO [32].

## 6. Previous Studies Analysis

Regarding the impact on the global tourism market, taking into account the possibility of the outbreak of a pandemic, most research to date has focused on individual countries or regions and just a few on a global level.

Cevik [33] tried to explain how and on what level infectious diseases affect the global tourism market. In this econometric approach, he created a gravity model of bilateral tourism flows among many countries, using previous infectious disease crises. The empirical analysis provides strong evidence that the international tourism market is negatively affected by the risk of infectious diseases, as measured by the number of confirmed cases in past episodes. He observes that, in the case of SARS, a 10% increase in the number of confirmed cases leads to a reduction of 9% in international tourist arrivals.

Using annual data and the PSVAR auto-regression model for 185 countries from 1995 to 2019 and system dynamic modeling with parameters connected to COVID-19, Marinko et. al. [34] estimated the potential impact of the current pandemic crisis on the global tourism industry. This study explained that pandemic crises have long-lasting negative effects on the global tourism industry and economy. They conclude that the estimated negative effects are far beyond those observed during other pandemic crises.

Kan et al. [35] studied the determinants and drivers of Hong Kong's inbound tourism, using data from January 2019 to December 2020. Five of Hong Kong's major regional tourism markets were selected: Japan, Malaysia, the Philippines, Singapore, and South Korea. Their empirical results using the 2-SLS method suggest that the COVID-19 pandemic has had significant negative impacts on the local tourism and aviation sectors and that such effects were asymmetric for the source and destination countries.

On the other hand, Farzanegan et al. [36] used data from a sample of more than 90 countries using multiple regression analysis. This resulted in a positive and significant association between the past records of international tourism and the cumulated numbers of confirmed cases and deaths resulting from COVID-19. Their finding was that international tourism suffered serious consequences from the COVID-19 outbreak.

## 7. Data

The data for this study were drawn from the databases of the World Bank, the OECD, the WHO, and Github, for the year 2020. Due to the limited availability of recent tourism

data, no panel data were used, but only cross-sectional data for the year 2020, in order to examine the impact of COVID-19, which emerged in late 2019. We tried to examine data for 228 countries, but due to the statistical significance of the sample, we used the 38 countries that offered data available on the vaccination against COVID-19 for 2020.

For tourism, the international tourism expenditure variable for travel goods at current prices (USD) is used, which is defined as the expenditure of international visitors, outbound to other countries. Specifically, it concerns the goods and services purchased by, or on behalf of, the traveler. Goods and services provided free of charge are not included. These expenditures also include same-day trips to other countries. The cost of air transportation of passengers is excluded; however, their goods and services are counted [37].

Income in this model is approximated by the variable GDP PPP, defined as the following: a country's GDP is the sum of the gross value added by all producers of goods and services residing in a country plus product taxes and minus subsidies that are not included in the value of products. Per capita values for gross domestic product (GDP) expressed in current international dollars are converted using a purchasing power parity (PPP) conversion factor. The conversion factor (PPP) is a spatial price deflator and currency converter that controls for price-level differences between countries [37].

I the income equation, net foreign direct investment outflows are used as an explanatory variable. Foreign direct investment refers to direct investment flows of capital share into an economy. Direct investment is a category of cross-border investment linked to a resident of an economy who has control or a significant degree of influence over the management of an enterprise company in another economy. Holding 10% or more of the common shares with voting rights is the criterion for determining the existence of a direct investment relationship [37].

The World Pandemic Uncertainty Index is used as an instrumental variable in our system of equations. The WPUI index is constructed to count the number of times the word "uncertainty" is mentioned near a word related to pandemics. Specifically, the index is the percentage of the word "uncertain" that appears near words related to a pandemic in the reports made by the country, and is multiplied by 1000. The higher its value, the greater the uncertainty resulting from pandemics, and vice versa [38].

Regarding the COVID-19 equation the variables used were the number of tests per case, the total number of vaccinations, and the total number of cases per million.

## 8. Methodology

The purpose of this study is to investigate the effects of COVID-19 on tourism in the countries which, by the end of 2020, had introduced a vaccination against COVID-19. Given the assumption that the pandemic caused by COVID-19 affects tourism not only directly but also indirectly, and based on its impact on income, the resulting system consists of the tourism (1), income (2) and COVID Equation (3): Table 1 presents the equations, while Table 2 presents, the variables used in the model.

**Table 1.** Definition of Equations.

| | | |
|---|---|---|
| Tourism equation: | $\text{Tourism} = \beta_0 + \beta_1 \text{Income}_i + \beta_2 \text{COVID}_i$ | (1) |
| Income equation: | $\text{Income}_i = k_0 + k_1 \text{COVID}_i + k_2 X_{1i}$ | (2) |
| COVID-19 equation: | $\text{COVID}_i = \lambda_0 + \lambda_1 \text{Vaccination}_i + \beta_3 X_{2i}$ | (3) |

where i denotes the respective country.

The variables used in the model are the following:

**Table 2.** Definition of Variables.

| | | |
|---|---|---|
| Tourism: | International tourism, expenditure for travel items (current USD) | -TOUR_EXP |
| Income: | GDP, PPP (current international USD) | -GDPPPP |
| COVID: | Total cases per million | -TOTALCASES |
| Vaccination against COVID: | Total vaccinations | -TOT_VAC |
| X1: | Foreign direct investment, net outflows (BoP, current USD) | -FDI_OUT_CUR |
| X2: | Tests per case | -TESTSPERCASE |
| IV List for 3SLS: | Total vaccinations, WPUI, tests per case | |

It is noted that in the tourism Equation (1), tourism is expected to be positively affected by income, as increasing purchasing power increases the possibility of making additional tourism expenditures. In addition, tourism is expected to be negatively affected by the spread of COVID-19, as tourist flows tend to decrease in cases of a pandemic, either due to concern about possible illness or due to travel restrictions on cross-border movements. This is the direct effect of COVID-19 on tourism. However, the spread of COVID-19 also indirectly affects tourism, through the negative effect it has on the economy and thus on the purchasing power of travelers.

The spread of COVID-19 is expected to be negatively affected by the performance of diagnostic tests (PCR, rapid), but also by vaccination, which was introduced late in 2020, although, as far as vaccination is concerned, its effectiveness as a measure to suppress the spread of COVID-19 may not have been seen immediately. According to the official UK government website, the first person worldwide to be vaccinated against COVID-19 was on 8 December 2020.

Regarding the variable of net foreign direct investment outflows, FDI_OUT_CUR is expected to have a positive effect on income (GDP, PPP (current international USD)) as it approaches investments.

Estimating a system of equations using the OLS method leads to specification error, due to the correlation between the disturbance terms of the structural equations and the independent variables. To avoid this problem, estimation methods such as "2SLS, 3SLS, SURE and FIML are used" [39], because in the case of autocorrelation we can erase it using Durbin's method by applying SURE/AR. Therefore, estimating a system of equations by the method of least squares is not appropriate, and thus estimation will be attempted initially with the SURE method. Of course, in the case of a problem—mainly heteroskedasticity—SURE is not suitable for estimating the system of equations, either. According to Hayashi [40], the 3SLS method provides the solution. It extends the systems of seemingly unrelated regressions, and it is also an appropriate method when the explanatory variables are correlated with the disturbance term and the residuals are characterized by heteroscedasticity and autocorrelation.

To use 3SLS, an instrumental variable needs to be included in the first stage of the equation, and should not be related to the residuals but must be related to COVID-19 [39]. Vaccination against COVID-19 is an instrumental variable, as it only affects tourism through the negative impact it has on the transmission of COVID-19.

The method of instrumental variables is used when it is suspected that one or more of the explanatory variables are characterized by endogeneity, that is, when the variable is correlated with the disturbance term [41].

## 9. Empirical Results

This study initially presents the estimated results for 38 countries which, at the end of 2020, had sufficient data from vaccination against COVID-19. An attempt was made to include variables such as government spending on education, ICU bed admissions, mask use, the Stringency Index, population density, inflation, unemployment and urbanization rate in the model equations, but they were not statistically significant and were all rejected. Table 3 presents the system estimation results with OLS, SURE and 3SLS.

**Table 3.** System estimation results with OLS, SURE and 3SLS.

| System Estimation | OLS | SURE | 3SLS |
|---|---|---|---|
| TOUR_EXP = C(1) + C(2) × GDPPPP + C(3) × TOTALCASES | | | |
| N | 37 | 37 | 31 |
| R-sq | 0.779380 | 0.760429 | 0.762787 |
| C | $7.72 \times 10^9$ ** | $1.36 \times 10^{10}$ *** | $1.77 \times 10^{10}$ *** |
| GDPPPP | 0.003739 *** | 0.003472 *** | 0.003546 *** |
| TOTALCASES | −172,676.9 * | −331,389.6 *** | −452,983.2 *** |
| GDPPPP = C(5) + C(6) × TOTALCASES + C(11) × FDI_OUT_CUR | | | |
| N | 37 | 37 | 31 |
| R-sq | 0.477047 | 0.472351 | 0.557785 |
| C | $3.02 \times 10^{12}$ ** | $3.66 \times 10^{12}$ *** | $4.07 \times 10^{12}$ *** |
| TOTALCASES | −64,814,983 * | −84,344,216 ** | $-1.15 \times 10^8$ *** |
| FDI_OUT_CUR | 48.08748 *** | 48.90300 *** | 91.65377 *** |
| TOTALCASES = C(8) +C(9) × TESTSPERCASE + C(10) × TOT_VAC | | | |
| N | 38 | 38 | 31 |
| R-sq | 0.150963 | 0.105620 | 0.163760 |
| C | 33,460.77 *** | 33,667.95 *** | 32,785.50 *** |
| TESTSPERCASE | −0.129059 ** | −0.199374 *** | −0.192296 *** |
| TOT_VAC | 0.004934 | 0.007096 ** | 0.007679 *** |

Asterisks *, **, *** depict statistical significance at 10, 5 and 1% level, respectively.

First, we notice that from estimating the system of equations with OLS, the resulting model is the following (Table 4).

**Table 4.** System Estimation with OLS.

| |
|---|
| TOUR_EXP = 7,722,770,515.08 + 0.00373887361774 × GDPPPP − 172,676.896179 ×TOTALCASES |
| GDPPPP = 3.02102227575 $\times 10^{12}$ + −64,814,982.6916 × TOTALCASES + 48.0874832676 × FDI_OUT_CUR |
| TOTALCASES = 33,460.7657539 − 0.129058571303 × TESTSPERCASE + 0.00493395189311 × TOT_VAC |

Using OLS, the estimated equation of tourism shows that the dependent variable (TOUR_EXP) is positively correlated with the independent variable (GDPPPP) and negatively correlated with (TOTALCASES), as expected. An increase in GDP is expected to positively affect tourism expenditure, as well as an increase in COVID-19 cases (TOTALCASES) leading to a reduction in tourist flows and thus in spending. The estimated coefficient of GDPPPP shows that a unit increase in GDPPPP causes TOUR_EXP to increase by 0.003739, while maintaining TOTALCASES constant. Similarly, the estimated coefficient of TOTALCASES shows that a unit increase in TOTALCASES leads TOUR_EXP to decrease by 172,676.9 while holding GDPPPP constant. Finally, when TOTALCASES = GDPPPP = 0, then TOUR_EXP = $7.72 \times 10^9$.

Next, observing the income equation, the variable for COVID-19, TOTALCASES, which represents the total cases per million, appears to have a negative effect on the income and thus the purchasing power of travelers, as opposed to an increase in the net FDI outflow variable (FDI), which leads to an increase in income.

Checking the statistical significance of the variables for the statistical significance level $\alpha$ = 0.10, we reject the null hypothesis, and the variables of the model are statistically significant, except for the vaccination variable, which is not statistically significant. Also,

for the first equation, the coefficient of determination equals 77.9%, while the adjusted R-squared equals 76.6%.

However, as mentioned above, this method does not give the best estimators when it comes to a system of equations and it is considered more appropriate to estimate with the SURE method, based on which the model is the following (Table 5).

**Table 5.** System estimation results with SURE.

| |
|---|
| TOUR_EXP = 13,614,794,055.4 + 0.00347210836767 × GDPPPP – 331,389.598754 × TOTALCASES |
| GDPPPP = 3.66275411847 × $10^{12}$ + −84,344,216.3041 × TOTALCASES + 48.9030023806 × FDI_OUT_CUR |
| TOTALCASES = 33,667.9495065 − 0.199373682529 × TESTSPERCASE + 0.00709649629471 × TOT_VAC |

The results of this method are improved in terms of statistical significance, as the standard errors using SURE are smaller than those obtained using OLS. However, SURE is also not suitable for estimating the specific system of equations, since we estimate each equation of the model separately with OLS. Also, we notice that for the first two equations we have a heteroskedasticity problem, as the *p*-value is smaller than the level of statistical significance for all commonly used levels ($\alpha$ = 1, 5, 10%) for the null hypothesis (Ho: no heteroscedasticity). However, no autocorrelation problem is observed (Table 6).

**Table 6.** Diagnostic results of Equations (1)–(3).

| Prob. F | Breusch–Godfrey Serial Correlation LM Test (Autocorrelation) | Heteroskedasticity Test: Breusch–Pagan–Godfrey |
|---|---|---|
| Tourism Equation | 0.4244 | 0.0000 |
| Income Equation | 0.6682 | 0.0000 |
| Covid Equation | 0.3406 | 0.4339 |

As mentioned above, the most appropriate method for the case of system estimation with heteroskedasticity is 3SLS, where a list of IVs is used, such as vaccination against COVID-19, the number of diagnostics for COVID-19 (test per case) and the World Pandemic Uncertainty Index (WPUI), as all these variables affect tourism only through the negative impact they have on the transmission of COVID-19.

The resulting model using 3SLS is the following (Table 7).

**Table 7.** System estimation with 3SLS.

| |
|---|
| TOUR_EXP = 17,748,666,060.6 + 0.00354631656905 × GDPPPP – 452,983.17832 × TOTALCASES |
| GDPPPP = 4.06834883466 × $10^{12}$ + −114,556,621.155 × TOTALCASES + 91.6537683406 × FDI_OUT_CUR |
| TOTALCASES = 32,785.4999822 − 0.19229583254 × TESTSPERCASE + 0.00767880183906 × TOT_VAC |

Estimating the system with 3SLS, we observe the preservation of the signs in all variables in relation to the results of OLS and SURE, although for all variables the statistical significance has improved. The negative direct and indirect effect of COVID-19 on the tourism sector is also observed.

The overall effect of COVID on tourism is expressed by the following equation:

$$COVID_i = k_1 \times \beta_1 + \beta_2 = (−114,556,621.155) \times (0.00354631656905) + (−452,983.17832) = −859,237$$

A unit increase in the COVID-19 approach variable (TOTALCASES) reduces international tourism expenditures by USD 859,237 for the 31 countries where 3SLS was finally implemented.

A concern arises regarding the fact that while the effect of vaccination on COVID-19 is expected to be negative, this does not appear in the results. This may have happened perhaps due to a lack of data, because the vaccination was actually introduced in December 2020 in the first countries and at least 3 weeks had to pass until partial immunity occurred, according to the pharmaceutical companies. Therefore, the results relate to a few countries and for a limited period of time; for this reason, no clear conclusions about vaccination were drawn.

## 10. Conclusions

In the present study an attempt was made to estimate the impact of the spread of COVID-19 on the global tourism market for 38 countries, for which there were enough data for vaccination at the end of 2020. The purpose of this study was to investigate whether tourism has been negatively affected by the spread of COVID-19 in recent years in terms of tourist flows and therefore tourist expenditures, firstly due to the increase in the concern of falling sick from COVID-19 after or during a trip, and secondly due to its effect on the income of tourists through the channel of the spread of COVID-19 in the economy.

In order to include both of the aforementioned effects of COVID-19 on tourism, a system of three equations was created, which was then estimated with the SURE method and then with 3SLS, due to the existence of heteroscedasticity. These three equations represent the direct impact of COVID-19 on tourism, the impact of COVID-19 on income, and the factors that affect the spread of COVID-19, such as diagnostic tests and vaccination. From the estimation of the system of equations, it is observed that COVID-19 has a negative effect on tourism through the two aforementioned channels.

The choice of this specific sample and year was made due to the need to investigate the effect from the moment the vaccination for COVID-19 began. Ideally, panel data would have been used for at least two years (2020–2021), but due to the limited availability of tourism data for the year 2021, cross-sectional data were used for 38 countries that introduced vaccination, and there were plenty of data up until the end of 2020. Also, an attempt was made to include variables such as government spending on education, ICU bed admissions, mask use, the Stringency Index, population density, inflation, unemployment and urbanization rate in the model equations, but they were not statistically significant and were ultimately rejected.

With the use of 3SLS, an equation was developed, with which we were able to calculate the direct and indirect impact of COVID-19 on tourism. More specifically, the dependent variable (TOUR_EXP) is positively correlated with the independent variable (GDPPPP) and negatively correlated with (TOTALCASES). As a result, we were able to calculate that a unit increase in the COVID-19 approach variable (TOTALCASES) reduces international tourism expenditure by USD 859,237.

However, a concern arises regarding the effect of vaccination on COVID-19, which, while expected to be negative, is not confirmed by the results, perhaps due to insufficient data. Therefore, the results were exported on the basis of limited data, as few countries had introduced vaccination programs by the end of 2020, and for a limited period of time, due to the unavailability of recent data on tourism.

Finally, the 3sls model that was developed in this specific research offers the opportunity for an in-depth analysis of the economic impact of various crisis events in other economic sectors apart from the global tourism market, such as the food industry, accommodation, the cruise sector, etc.

**Author Contributions:** Conceptualization, I.-P.V. and T.M.; methodology, I.-P.V.; software, I.-P.V.; investigation, I.-P.V.; resources, I.-P.V.; data curation, I.-P.V.; writing—original draft preparation, I.-P.V.; writing—review and editing, I.-P.V. and T.M.; supervision, T.M.; project administration, T.M. All authors have read and agreed to the published version of the manuscript.

**Funding:** This research received no external funding.

**Institutional Review Board Statement:** Not applicable.

**Informed Consent Statement:** Not applicable.

**Data Availability Statement:** All resources are provided at the references sector.

**Conflicts of Interest:** The authors declare no conflicts of interest.

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
