# Peer review of "Pandemic and Economy: An Econometric Analysis Investigating the Impact of COVID-19 on the Global Tourism Market"

_tourismhosp, doi:10.3390/tourhosp5020019_

Round 1
Reviewer 1 Report
Comments and Suggestions for Authors
Dear Authors,
The issue of the negative impact of the COVID-19 pandemic on the tourism economy has been studied many times and is richly reflected in scientific literature. It was a terrible disease, but now, in 2024, the situation is different. The crisis is over and tourism has come back to life. It seems to me that it would be quite a challenge now to write a groundbreaking scientific article on the issues of the COVID-19 pandemic. It is a pity that the empirical research covers only two years. I wonder about the topicality of your article, especially since some online sources come from 2020.
You wrote that “the purpose of the research was to investigate the negative effects of COVID-19 on the global tourism market for the year 2020-2021”. Therefore, while reading your article, I was wondering what new things your research will bring to theory and practice. You write that previous research has covered smaller areas, while yours is global. I have doubts about whether the analysis of only 38 countries allows us to write about the global dimension.
The information you provided in the abstract is not precise for me. It is worth writing more precisely in the abstract what the purpose of your research was, what methods you used, and what results you obtained. The reader will be interested in conclusions and recommendations for the future. Your current conclusions at the end of the article are disappointing.
In the theoretical part, you rightly show the events that negatively affected air transport and tourist traffic. This is illustrated in Figure No. 1. It is a pity that the analyzed period ends in 2018. It would be interesting for the article to show data from later years.
I don't like the beginning of chapter II. In lines 57-58, it is unnecessary to mention that the goal of tourism is "participation in different traditions". It is a truism to say that "tourism is often combined with activities other than pleasure". I would also omit controversial considerations when the phenomenon of tourism appears. After all, man has been traveling for various purposes since ancient times, and the development of transport has only made tourism more dynamic.
It is worth refining the technical side of the article, including unifying the graphics. For example, A4 and A5 charts are in a completely different style. Why can't figures and tables have uniform numbering?
Author Response
Dear Reviewer
Thank you for your valuable comments
All the best
Authors

Reviewer 2 Report
Comments and Suggestions for Authors
Dear authors,
the paper is interesting and so is the topic.
For the publisher MDPI, several studies have also been published in other magazines analyzing the impacts of Covid on tourism. For example "A Review of Research on Tourism Industry, Economic Crisis and Mitigation Process of the Loss: Analysis on Pre, During and Post Pandemic Situation".
However, some corrections need to be made.
1) Lines 269-270: The he sample consists of 38 countries and was determined by the data available on the vaccination against Covid-19 for 2020. Please, justify the statistical significance of the sample according to the literature.
2) What is the novelty related to this study?
3) Is a study replicable considering other industries (e.g. food and catering, restaurant)?
4) Please, insert a section of limitation in Conclusion paragraph.
5) Please, insert the loss data due to the pandemic for example for the cruise sector.
6) Please, consult other literature products.
Author Response

(The authors gave the same response as above.)

Round 2
Reviewer 1 Report
Comments and Suggestions for Authors
Dear Authors,
I see that some of the comments have been taken into account. It's a pity that the graphics couldn't be unified. Figure 1 shows 2 variables, while the description contains one variable. The inscription on the right side of the drawing requires a space. I still don't understand the numbering concept. Why is one drawing number 1 and the next one number A2.
Author Response

(The authors gave the same response as above.)

Reviewer 2 Report
Comments and Suggestions for Authors
Dear authors,
thanks for the improvement carried out on the manuscript.
1) Abstract: it is ok, adding also the sentence "Using 3SLS, an equation was developed to calculate the direct and indirect impact of Covid-19 on tourism. Notably, the dependent variable (tourism expenditures) exhibited a positive correlation with the independent variable (GDP) and a negative correlation with the total Covid-19 cases. Consequently, it was determined that a unit increase in the Covid-19 variable led to a reduction of international tourism expenditures by US$859,237.
2) Data section: it is ok, adding also "We tried to examine data for 228
countries, but due to statistical significance of the sample, we used the 38 countries that offered data available on the vaccination against Covid-19 for 2020.
3) Table A3: please, reduce the size.
4) Conclusion: it is ok, adding also several improvements amd sentences.
5) Please, insert a brief note of the novelty related to the study and the methodology used.
6) Is it a replicable study? Please, motivate.
Author Response

(The authors gave the same response as above.)
